# Macroprudential policies and CO2 emissions: A comparative analysis of G7 and BRIC countries

Heng Luo [1,2]*, Fakarudin Kamarudin[2]

1 School of Digital Economy and Industry, Jiangxi University of Engineering, Xinyu, Jiangxi, China, 2 School of Business & Economics, Universiti Putra Malaysia, Serdang, Malaysia

* barryluoheng@foxmail.com

## Abstract

This study investigates the impact of macroprudential policies on CO2 emissions in G7 and BRIC countries using country-level panel data from 11 countries, covering the period from 1992 to 2020. The findings indicate that macroprudential policies alleviate CO2 emissions in the sample. Quantile regression results reveal that policies can exacerbate CO2 emissions in countries with high levels of CO2 emissions due to carbon leakage. The positive impact of macroprudential policies on sustainable development can be strengthened by high level of globalisation. Moreover, the influence of macroprudential policies stayed the same based on the basic regression results during the post-global financial crisis (GFC) period, while the impact was positive in the pre-GFC period. Finally, robust tests validated the findings reported in the basic regression model. From this, policymakers should prioritise sustainable economic growth when implementing macroprudential policies and leverage the influence of globalisation to amplify their impact on CO2 emissions. Furthermore, it is crucial to strengthen environmental regulations to prevent carbon leakage that result from industries seeking lenient standards.

**Data Availability Statement:** The data about macroprudential policies can be found in https://www.elibrary-areaer.imf.org/Macroprudential/Pages/Home.aspx, other variables can be found in World Development Indicators.

## 1. Introduction

### 1.1 Background and gap

The escalating levels of CO2 emissions have contributed to widespread environmental degradation and climate change across the globe, leading to substantial adverse impacts on the health of the ecosystem that contributed to human sickness, droughts, and floods. To alleviate the negative impact of CO2 emissions, many scholars have examined numerous socioeconomic, political, and environmental aspects that can impact emissions [1–4]. Existing research has extensively examined the key factors influencing CO2 emissions, including ICT, economic growth, economic uncertainty, financial inclusion, natural resource rents, and renewable energy consumption [2, 5–7]. One of the most influential factors that affect emissions is macroprudential policies, which is notably significant and should not be overlooked [8, 9].

**Funding:** The author(s) received no specific funding for this work.

**Competing interests:** The authors have declared that no competing interests exist.

There are several explanations to the negligible role of macroprudential policies on CO2 emission. In principle, macroprudential policies have a dual-edged impact on CO2 emissions, where tools like the Loan-to-Value (LTV) could, for instance, lead to a reduction in economic activity. However, while the resulting slower pace of economic growth may appear detrimental, it often leads to decreased energy consumption, reduced transportation demands, and a drop in industrial production that collectively contribute to a reduction in CO2 emissions. Yet, on the flipside, macroprudential measures are instrumental in ensuring the stability of the financial system [10], which in turn can lead to increased economic activities that result in higher emissions. One issue that needs to be highlighted, however, is that there is a lack of research on whether the impact of macroprudential policies on CO2 emissions varies across countries with different levels of CO2 emissions' intensity and different level of globalisation.

In contemporary economics, achieving a greater gross domestic product (GDP) is essential. Both BRIC countries (Brazil, Russia, India, China) and G7 countries (Canada, France, Germany, Italy, Japan, the United Kingdom, and the United States) in particular are known to have achieved substantial amounts of GDP. However, as explained earlier, the attainment of such significant GDP levels comes at the cost of consuming a vast amount of fossil fuels and emitting a substantial quantity of CO2. Consequently, these nations face the pressing challenge of addressing the critical issue of high-volume carbon dioxide emissions. As depicted in Fig 1, BRIC and G7 countries collectively account for 45.22% and 21.78% of the total CO2 emissions in the year 2020.

## 1.2 Objective of the study and reasons for sample selection

Following above, to cater to the identified gap in research, this study sought to determine whether macroprudential policies impose a positive or negative effect on CO2 emissions. In addition, this research attempts to analyse the impact of macroprudential policies on CO2 emissions in G7 and BRIC countries. There are three factors that influenced the sample selection. First, in terms of economic scale, these countries are the top eleven countries ranked by

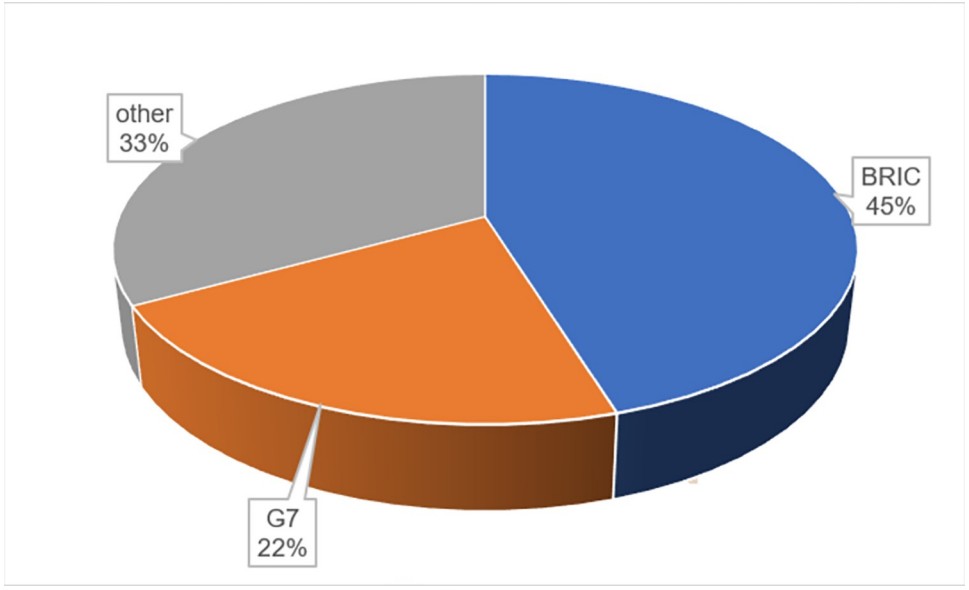

**Fig 1. Share of BRIC and G7 countries in global CO2 emissions (kt) in 2020.** (Data Source: World Development Indicators (WDI)).

GDP based on the data from the World Bank statistics for 2019. These countries are divided into two groups: the BRIC groups, which are made up of Brazil, Russia, India, and China, and the G7 groups, which are represented by Canada, France, Germany, Italy, Japan, the United Kingdom, and the United States. Second, among top annual CO2 emitting countries 2020 (based on data for fossil fuels), these 11 countries were identified as important contributors. For example, within these countries, China alone is responsible for 31% of total CO2 emissions while India, Brazil, Russia, Canada, France, Germany, Italy, Japan, the United Kingdom, and the United States are responsible for 7%, 1%, 5%, 2%, 1%, 2%, 1%, 3%, 1% and 13% respectively(https://www.ucsusa.org/resources/each-countrys-share-co2-emissions). The data reveals that approximately 67% of global carbon emissions can be attributed to these 11 countries. This underscores the substantial influence that these economies wield on climate change, global warming, and environmental degradation. Third, these countries are parties to both the Kyoto Protocol and the Paris Agreement. They have made their commitment to address global environmental issues, particularly climate change, and to reduce or mitigate greenhouse gas emissions. Additionally, the ongoing development of the global value chain is another issue, which should not be neglected in the factors concerning environmental quality [11].

## 1.3 Findings and implications

The regression results in this study indicate that macroprudential policies reduce CO2 emissions. Quantile regression results indicate that macroprudential policies tend to alleviate CO2 emissions in countries with relatively low levels of CO2 emissions while deepening CO2 emissions in countries with high levels of CO2 emissions. A higher level of globalisation will also amplify the positive effect of macroprudential policies on environmental quality. Specifically, a positive relationship was observed during the pre-GFC period, while the relationship was consistent with the basic regression results in the post-GFC period. The robustness tests further confirmed these findings, supporting the robustness of our results. Sustainable economic growth should be prioritised when implementing macroprudential policies. Additionally, environmental regulations should be strengthened to prevent carbon leakage.

## 1.4 Contributions

There are three key novelties of this study when compared with the previous literature. First, from a theoretical point, the positive role that macroprudential policies play in combating environmental degradation was strengthened. While most prior research has predominantly focused on examining the influence of various factors such as ICT [1, 12, 13], financial development [1, 2], financial inclusion [6, 14, 15], and renewable energy consumption [2, 16] on CO2 emissions, this study considered the carbon reduction effect of macroprudential policies. Hence, this paper fills the gap by assuming that these policies can contribute to Sustainable Development Goals (SDGs) by influencing credit and investment decisions. Second, from an empirical perspective, this study aimed to demonstrate the linear and non-linear effect of macroprudential policies on CO2 emissions. Multivariate Panel Regression Analysis (MPRA) and quantile regression was used on a panel data covering 11 countries from 1992 to 2020 to analyse the impact of macroprudential policies on CO2 emissions. In terms of methodology, the quantile regression model not only overcame the disadvantage of traditional methods of focusing on the conditional mean but also depicted the varying effect of macroprudential policies across the conditional distribution of CO2 emissions. Our empirical analysis revealed that macroprudential policies plunged CO2 emissions in countries with comparatively low CO2 emissions levels while the effect was the opposite in comparatively higher CO2 conditions. Third, globalisation was found in this study to be effective in promoting the impact of

macroprudential policies on environmental quality. This work focused on the mechanism by which the carbon reduction effect of macroprudential policies may be strengthened by globalisation, whereas the majority of the existing literature had neglected this point.

## 1.5 Structure of the paper

The remainder of the paper is organised as follows. Section 2 is the literature review. Section 3 describes the methodology. Section 4 reports the empirical results. The last section emphasises the key conclusions and implications for policymakers.

## 2. Literature review

### 2.1 Macroprudential policies-CO2 emissions nexus

The existing body of literature concerning macroprudential policies has predominantly concentrated on the nexus between these policies and the financial sector while paying scant attention to the connection between macroprudential policies and environmental quality. Tightening macroprudential policies, for instance, was found to reduce EDF and Z-score [17] and reduce bank risk-taking levels [18]. Although macroprudential instruments strengthen the stability of financial institutions, they can also restrict lending activity, particularly for smaller enterprises that rely largely on bank credit and have other limited sources of finance [19]. Loan expansion is limited by macroprudential policies, which bank state ownership alleviates [20]. However, these studies did not extend their analysis to examine how changes brought about by macroprudential regulations in financial sectors may affect CO2 emissions.

The literature concerning the relationship between macroprudential policies and environmental quality can be broadly categorised into two trends. One argues that macroprudential policies influence environmental quality through their impact on credit, while the other posits that they affect SDGs by directing investment. In terms of the impact of macroprudential policies on investment, some believe that macroprudential policies negatively affect green investment. For instance, policies implemented in the wake of the financial crisis, most notably Basel III, appeared to favour short-term rather than long-term green investments [21]. This point is in line with the studies of [22, 23], who theorised that macroprudential policies have incentivised short-term 'brown' investments at the expense of long-term, climate-friendly ones even though reduced green investments will lead to more CO2 emissions. Nevertheless, a positive relationship between macroprudential policies and green investment have also been documented in literature. A relatively simple mechanism is provided by maximum credit ceilings and minimum credit floors to direct investments towards "green" projects [24]. The former functions by limiting the maximum bank resources to carbon-intensive or polluting activities, while the latter allocates a minimum number of resources to green ones. The green supporting factor (GCF) works by assigning less risk weights to greener investments while the brown penalising factor (BPF) offers more risk weights to polluted ones [25]. This lessens the polluting activities while increasing green ones.

Concerning the positive impact of macroprudential policies on credit, differentiated reserve requirements have allocated lower reserves when dealing with green loans [8]. This allows banks to lend out more money, which increases their business volume and makes them more inclined to offer green credit [21]. Nevertheless, a negative correlation is also documented in the study of [9], who contended that financial institutions with a higher carbon-intensive credit may need to adopt the higher countercyclical capital buffer. This higher buffer requirement will discourage credit flow towards carbon-intensive industries.

Finally, some scholars have expressed doubts regarding the effectiveness and cost of macroprudential policies, as they tend to be more effective only in emerging economies compared to

advanced economies [26]. This is because emerging economies have greater flexibility in credit allocation and utilise a wider range of monetary policy instruments, whereas advanced economies primarily rely on interest rate adjustments. Hence, adequate calibration and timely activation is necessary for effective countercyclical capital buffers [25]. The cost of macroprudential policies on excessive green credit leads to an unfavourable trade-off between financial stability and environmental sustainability [8].

Based on the above discussion, it can be inferred that macroprudential policies exert an influence on CO2 emissions.

## 2.2 Globalisation-CO2 emissions nexus

Concerning the impact of globalisation on CO2 emissions, the conclusions of the researchers have been controversial thus far. For instance, a negative relationship is documented in the works of [27, 28], while a positive relationship is concluded in the research of [28–30] used ARDL and NARDL models to examine the nexus in MENA countries and concluded that financial globalisation cuts CO2 emissions through technology transfer and the expansion of their financial markets. Similarly, [30] used the ARDL to test the impact of globalisation on CO2 emissions in Turkey and concluded a positive relationship. This phenomenon can be attributed to the scale effect, which suggests that greater energy consumption is associated with larger volumes of foreign trade. [29] concluded that globalisation accelerates CO2 emissions in 78 developing economies from GMM results and that governments should enforce taxes.

Globalisation can influence the impact of macroprudential policies on CO2 emissions, where the moderating effect can function in different aspects. For instance, political globalisation can encourage nations to negotiate environmental policies and cooperate to reduce carbon emissions. As an example, the Paris climate agreement aims to ensure that global warming remains well below 2 degrees Celsius [31], while the Kyoto Protocol aims to cut their yearly GHG emissions by an average of 5.2%, which is below 1990 levels for the years 2008 to 2012 [32]. As a result, the Kyoto Protocol has been shown to effectively reduce CO2 emissions, as demonstrated by the difference-in-differences estimator [33].

Apart from political globalisation, cultural globalisation has also been found to lead to the spread of ideas, values, and norms related to environmental conservation [34]. As a result, more people may become aware of and concerned about environmental problems, such as CO2 emissions. This means that people and communities may be more likely to accept and adhere to policies that attempt to cut back on emissions. From this, it is likely that globalisation can therefore enhance the ability of macroprudential policies to reduce CO2 emissions. However, no literature to date has considered the moderating role of globalisation with macroprudential policies. Therefore, this study fills the gap and extends the frontier of knowledge by considering the moderating effect of globalisation on this nexus.

## 3. Data and methodology

The aim of this study is to analyse the influence of macroprudential policies on CO2 emissions of G7 and BRIC countries. The rationale behind selecting these countries is that they represent the top 11 nations ranked by GDP based on data from the World Bank in 2019. Data was gathered from a variety of sources. Firstly, CO2 emissions (kt) was chosen as the dependent variable. Secondly, the MaPR_3 index was utilised as a measure of the level of macroprudential policies, and for this purpose, relevant data was acquired from [35], who had compiled a comprehensive database based on annual records of macroprudential policies. The rationale behind selecting the period from 1992 to 2020 was due to the availability of macroprudential

policy records spanning from 1990 to 2021 and CO2 emissions records spanning until 2020. Moreover, constructing the MaPR_3 index required employing a 3-year rolling data as an indicator of macroprudential regulation in year "t", hence, the study period encompassed the years from 1992 onwards. Thirdly, the KOF globalisation index from the Swiss Economic Institute [36] was chosen. Finally, the country-level control variables such as GDP, ICT, foreign direct investment, CPI, and openness index were gathered from WDI.

## 3.1 Dependent variable: CO2 emissions

Adopting the approach used by [37–39], CO2 emissions (kt) sourced from WDI was chosen as the dependent variable.

## 3.2 Independent variable: macroprudential policies

Even before the breakout of the global financial crisis (GFC), monetary authorities and financial regulators in developing countries had actively devised and implemented a diverse array of macroprudential tools that had neither tightened nor loosened in recent years. These policies play a pivotal role in credit control that guide investment directions to ensure economic stability.

Macroprudential policies have been investigated in a wealth of studies [17, 40–43]. In extant literature, there are two main types of macroprudential policies; one is from [44], and the other is from [35]. This study chose the latter database due to its broader coverage of both countries and time periods.

The macroprudential tools data from [35] collected the history of a wide range of macroprudential instruments including 17 distinct measures on a monthly basis for the period spanning from 1990 to 2021. The researcher assigned a numerical value to each macroprudential measure, with a value of 1 representing a tightening tool, -1 indicating a loosening tool, and 0 representing otherwise. The aggregate index for each country during a specific month was calculated as the total sum of the numerical values assigned to the 17 macroprudential measures. For example, if one country (1) loosens the requirements for loan loss provision and limit on leverage of banks, (2) tightens limits to the loan-to-value ratios, and (3) maintains neutrality for the other macroprudential instruments in a given month, then the aggregate index for that country in that month will be -1–1 + 1 = -1. In this context, the index serves as a net value, reflecting the total sum of tightening and loosening tools for that country.

Although the aggregate index is accessible on a monthly basis, this study required annual variables. Hence, in this study, the aggregate index was added for each of the twelve months of a given year to generate yearly variables, which resulted in a single number that indicates the macroprudential policy stance for that specific year. A positive value denotes a policy position that is tighten-oriented for that year, whereas a negative value denotes a policy stance that is loosen-oriented. The research may examine longer-term patterns and shifts in macroprudential policy by using this annual aggregate.

It is unpredictable when these macroprudential regulations will impose effect on banks and borrowers [45]. Similarly, the implementation of macroprudential actions, such as changes in capital requirements, loan-to-value ratios, or reserve requirements, can have a delayed effect on CO2 emissions. This transmission lag refers to the time it takes for the impact of macroprudential policies to become fully evident on CO2 emissions. Furthermore, the effects of macroprudential policies often endure long after their implementation. Due to above, relying solely on the current influence of macroprudential policies may mean that their impact on CO2 emissions is overlooked. Following the method of [17, 40], this study takes a three-year aggregated value to measure the macroprudential policies in certain year. For example, the value of

this index of country in 2019, 2020, and 2021 is 1, 2, and 3 separately. Then, the actual index used for 2021 will be (1+2+3) 6. MaPP_3 was substituted for the real value of macroprudential regulation in year t. Because the database covers the period from 1990 to 2021, three-year aggregated values are needed to generate MaPP_3, where the time range for the MaPP_3 variable is from 1992 to 2021.

## 3.3 Moderating variable

The KOF globalisation index is an overall index that assesses the politics, business, and social aspects of globalisation [46]. This index varies from 1 to 100, with a larger value representing a higher degree of globalisation. Theoretically, globalisation facilitates the dissemination of advanced green technologies, which benefits the environment and reduces trade barriers, thus accelerating the circulation of green products. People can communicate more easily across boundaries as a result of globalisation [47]. In addition, cultural globalisation promotes environmental awareness, which contributes to sustainable development.

## 3.4 Control variable

Previous research has indicated that the external macroeconomic environment has an impact on CO2 emissions. For instance, ICT may impose stress on CO2 emissions because excessive energy consumption caused by the use of numerous inefficient ICT devices and short product life cycles of ICT equipment will emit more CO2 [1]. Economic growth, accompanied by an increase in energy consumption, has a significant impact on environmental pollution [12]. Lax environmental regulations in less developed nations attract foreign capital with high pollution levels that are looking for a 'pollution haven' to avoid the high costs of pollution control compliance [4]. Hence, high-income nations would have lower CO2 emissions as they stop manufacturing pollution-intensive commodities domestically and instead import them from other nations [48]. Finally, an increase in price levels may also erode the purchasing power of government spending, leading to reduced demand, which in turn can lead to improvements in environmental sustainability [49]. Based on the above discussion, the following variables were chosen: 1) ICT is measured by mobile subscriptions per 100 people. 2) Foreign direct investment (FDI) is the share of the net inflows of FDI of the GDP. 3) Openness index (trade) is measured by the ratio of the sum of total exports and total imports to GDP. 4) Inflation: consumer price index (CPI) (2010 = 100) is used to characterise it. 5) Economic growth: GDP per capita(log) is selected as the proxy of economic growth.

The data for the variables above are summarised in Table 1.

## 3.5 Econometric model

This step of this study was to find out the impact of macroprudential policies on ecological footprint by MPRA, which include ordinary least square (OLS), fixed effect model (FEM), and random effect model (REM). The Breusch Pagan (BP) and Lagrangian Multiplier (LM) tests formed the foremost step as these tests can detect whether pooled or panel data is optimal. If the p-value of the BP test and the chi-square of LM test is significant at 5% level, the panel data was chosen. Both FEM and REM were employed in this study to deal with panel data. The Hausman test was used to choose the suitable model for this research based on the null hypothesis. FEM was chosen to analyse the data if the null hypothesis was rejected (or when the prob.$< 0.05$). Hence:

H0: The random effect is appropriate

**Table 1. Variables explanation.**

|  | Variable | Symbol | Description | source |
|---|---|---|---|---|
| Dependent variable | CO2 emissions | CO2 | CO2 emissions (kt)(log.) | WDI |
| Independent variable | macroprudential policies | MaPP_3 | Measure the level of macroprudential policies | [35] (https://www.elibrary-areaer.imf.org/Macroprudential/Pages/Home.aspx) |
| Control variables | ICT | ICT | Mobile subscriptions per 100 people | WDI |
|  | Foreign direct investment | fdi | Foreign direct investment, net inflows (% of GDP) | WDI |
|  | Openness index | trade | total exports + total imports (% of GDP) | WDI |
|  | Inflation | CPI | Consumer price index (2010 = 100) | WDI |
|  | Economic growth | GDP | GDP per capita(log) | WDI |
| Moderator variable | Globalization | KOF | Globalization index | Swiss Economic Institute [36] |

H1: The random effect is not appropriate. The following empirical equation was proposed:

$$CO2_{it} = \alpha_0 + \alpha_1 \text{MaPP\_3}_{it} + \sum_{a=1}^{4} \beta_a CC_{it} + \varepsilon_{it}$$

$$CO2_{it} = \alpha_0 + \alpha_1 \text{MaPP\_3}_{it} + \alpha_2 \text{KOF}_{it} + \alpha_3 \text{MaPP\_3}_{it} * \text{KOF}_{it} + \sum_{a=1}^{4} \beta_a CC_{it} + \varepsilon_{it}$$

where:
$CO2_{it}$ = the log term of CO2 emissions (kt) of country i at time t
$\text{MaPP\_3}_{it}$ = macroprudential policies of country i at time t
$\text{KOF}_{it}$ = globalisation index of country i at time t
$CC_{it}$ = country-level control variable of country i at time t
$\varepsilon_{ijt}$ = the error term

## 4. Empirical results

### 4.1 Descriptive statistics

Table 2 presents the descriptive statistics of variables in the sample.

In terms of the dependent variable, the mean value of CO2 emissions was 13.78 with a standard deviation of 0.995. However, the G7 countries scored below average (13.56), while the BRIC scored above average (14.17). In the sample, the independent variable's mean value was 3.79. In particular, the mean value of macroprudential policies in BRIC countries was 6.793, compared to 2.074 in G7 groups. This result is in line with the work of [44], who indicated that emerging market economies rely heavily on policies in contrast to developed economies. Significant fluctuations around the sample mean can also be observed for other control variables.

The VIF value in Table 3 indicates that there was no significant multicollinearity among the variables in the regression model since the VIF value for each variable was less than 10.

### 4.2. Basic results

In Table 5, MaPP_3 (macroprudential policies) and control variables were included in the model. In the preliminary stage, the results from Table 5 show that the fixed effect model was most suitable to be used in this study because the p value of BP test and Chi-square of LM test

**Table 2. Descriptive statistics.**

|  | variable | N | mean | p25 | p75 | sd |
|---|---|---|---|---|---|---|
| CO2 | ALL | 319 | 13.78 | 12.98 | 14.31 | 0.995 |
|  | BRIC | 116 | 14.17 | 13.25 | 14.71 | 1.064 |
|  | G7 | 203 | 13.56 | 12.93 | 13.96 | 0.882 |
| MaPP 3 | ALL | 319 | 3.79 | 0 | 6 | 5.496 |
|  | BRIC | 116 | 6.793 | 1 | 11 | 6.587 |
|  | G7 | 203 | 2.074 | 0 | 2 | 3.837 |
| trade | ALL | 319 | 46.02 | 29.51 | 58.47 | 18.01 |
|  | BRIC | 116 | 39.73 | 26.07 | 49.86 | 15.38 |
|  | G7 | 203 | 49.62 | 32.82 | 61.85 | 18.45 |
| fdi | ALL | 319 | 2.11 | 0.765 | 2.991 | 1.911 |
|  | BRIC | 116 | 2.341 | 1.044 | 3.542 | 1.44 |
|  | G7 | 203 | 1.977 | 0.602 | 2.537 | 2.126 |
| CPI | ALL | 319 | 91.49 | 77.71 | 106.7 | 30.5 |
|  | BRIC | 116 | 86.47 | 51.65 | 118.2 | 46.96 |
|  | G7 | 203 | 94.35 | 82.72 | 105.5 | 13.66 |
| ICT | ALL | 319 | 68.5 | 14.04 | 108.7 | 50.56 |
|  | BRIC | 116 | 55.56 | 1.344 | 96.95 | 54.75 |
|  | G7 | 203 | 75.9 | 30.82 | 113.3 | 46.55 |
| GDP | ALL | 319 | 9.697 | 8.927 | 10.57 | 1.243 |
|  | BRIC | 116 | 8.24 | 7.425 | 9.026 | 0.92 |
|  | G7 | 203 | 10.53 | 10.38 | 10.65 | 0.181 |

was significant at 1% level or lower and the p value of Hausman test was significant at 1% level or lower.

The fixed effect model in Table 5 shows that MaPP_3 had a significant and negative effect on CO2 emissions. Hence, it can be concluded that there is a negative association between the use of macroprudential policies and CO2 emissions. This relationship can be explained by the impact of these policies on credit and investment decisions. On one hand, macroprudential policies can curb credit expansion [44]. For instance, financial institutions with a higher carbon-intensive credit may need to adopt a higher countercyclical capital buffer [9]. Therefore, banks will offer less credit to the carbon-intensive industry. Reduced credit may result in the carbon-intensive decreased borrowing and expenditure by families and enterprises [50]. Reduced activities of families and enterprises, in turn, will affect economic activities that are

**Table 3. VIF.**

| Variable | VIF | 1/VIF |
|---|---|---|
| CPI | 2.39 | 0.419 |
| ICT | 2.32 | 0.431 |
| MaPP 3 | 1.67 | 0.597 |
| GDP | 1.54 | 0.649 |
| trade | 1.24 | 0.809 |
| fdi | 1.06 | 0.943 |
| Mean VIF |  | 1.7 |

The correlation coefficients were not high as shown in Table 4, suggesting that multicollinearity may not exist.

**Table 4. Correlation matrix.**

|  | CO2 | MaPP 3 | trade | fdi | CPI | ICT | GDP |
|---|---|---|---|---|---|---|---|
| CO2 | 1 |  |  |  |  |  |  |
| MaPP 3 | 0.306*** | 1 |  |  |  |  |  |
| trade | -0.257*** | 0.114** | 1 |  |  |  |  |
| fdi | -0.014 | 0.094* | 0.213*** | 1 |  |  |  |
| CPI | 0.099* | 0.478*** | 0.089 | 0.073 | 1 |  |  |
| ICT | -0.057 | 0.350*** | 0.268*** | 0.058 | 0.702*** | 1 |  |
| GDP | -0.229*** | -0.246*** | 0.266*** | -0.012 | 0.233*** | 0.368*** | 1 |

energy-intensive and contribute to CO2 emissions. For example, industries may cut back on production, leading to lower energy consumption and emissions.

[21] theorised that low-carbon activities will also benefit from macroprudential policies. On the other hand, these policies can counter the housing bubble [51], which may influence urban development patterns. From this, reduced real estate projects may also lead to more compact and energy-efficient urban designs, which can reduce emissions associated with transportation and infrastructure development.

Finally, macroprudential policies have the potential to steer investment in specific directions. For instance, Bank Indonesia has introduced macroprudential measures including

**Table 5. Basic regression result.**

|  | ols | fe | re |
|---|---|---|---|
| MaPP_3 | 0.062*** | -0.008*** | -0.007*** |
|  | (0.01) | (0.00) | (0.00) |
| trade | -0.014*** | 0.002** | 0.002** |
|  | (0.00) | (0.00) | (0.00) |
| fdi | 0.006 | 0.012*** | 0.012*** |
|  | (0.03) | (0.00) | (0.00) |
| CPI | 0.002 | 0.002*** | 0.002*** |
|  | (0.00) | (0.00) | (0.00) |
| ICT | -0.003* | -0.003*** | -0.003*** |
|  | (0.00) | (0.00) | (0.00) |
| GDP | -0.033 | 0.955*** | 0.932*** |
|  | (0.05) | (0.04) | (0.04) |
| constant | 14.493*** | 4.480*** | 4.695*** |
|  | (0.47) | (0.37) | (0.51) |
| N | 319 | 319 | 319 |
| r2 | 0.194 | 0.753 |  |
| r2_a | 0.178 | 0.740 |  |
| F | 12.498*** | 153.522*** |  |
| BP LM | chibar2(01) = 3423.80*** | | |
| chi2 |  |  | 863.744*** |
| Hausman | | chi2(5) = 19.68*** | |

Standard errors in parentheses

* $p < 0.1$

** $p < 0.05$

*** $p < 0.01$

green LTV and green down payment to increase the amount of green real estate and green auto loans [52]. Additionally, other macroprudential instruments have been documented to influence investment decisions. Differentiated reserve ratio requirements in favour of the green also result in a larger amount of credit, incentivising financial institutions to allocate more resources towards environmentally-friendly projects as the primary revenue source for banks is lending [53]. [21] contend that banks might adjust the risk-weighted capital ratios when implementing green supporting factors so that low-carbon activities would have less impact on the balance sheet. As a result, banks will be encouraged to finance climate-related investments. More investments in green projects will cut down the CO2 emissions and improve environmental quality.

Regarding the control variables, the regression results show that ICT negatively contributes to CO2 emissions, while FDI, trade, GDP, and CPI have a positive impact on CO2 emissions. The results in this study indicate that ICT is negatively associated with CO2 emissions. This phenomenon can be explained by the fact that ICT promotes greening and raises environmental awareness, which both reduce emissions. This is consistent with the study of [54, 55].

Next, FDI has a significant positive influence on CO2 emissions. This is consistent with the findings of earlier research of [37]. Regarding the role of trade, the regression results indicate that improving the amount of trade can cause unfavourable environmental conditions. This phenomenon can be attributed to the escalation of production and transportation activities, which consequently demand greater energy consumption. The heightened consumption of fossil fuels as a result of these activities is also directly linked to an increase in CO2 emissions, where similar findings were reported in the study of [11, 15].

In terms of economic growth, the positive coefficient of GDP indicates that economic growth aggravated CO2 emissions. This conclusion aligns with the expectations of this study, where economic expansion significantly influences environmental pollution due to the concurrent rise in energy consumption. Hence, the findings of this study is in line with the work of [1].

Finally, CPI plays an important role in accelerating CO2 emissions, where an increase in the CPI may signify a healthy economy with a higher consumption pattern, which would then lead to more CO2 emissions due to the increased demand for products and services that need energy-intensive production methods.

## 4.3. Quantile regression

In line with the study of [56], we examined the impact of macroprudential policies on CO2 emissions at the 10th, 25th, 50th, 75th, and 90th quantiles. The results can be seen in Table 6. The coefficient was found to be negatively significant at lower quantiles (10th quantile), insignificant at 25th, and positively significant at higher quantiles from 50th. The results indicated that macroprudential policies plunged CO2 emissions in countries with comparatively low CO2 emissions levels. In addition, macroprudential measures had a pronounced detrimental impact on emissions in 10th quantile nations with lower CO2 emissions. The issue of carbon leakage was the main cause of this phenomenon. Hence, in this situation, macroprudential regulations served as lending restrictions. However, these constraints tend to disproportionately impact countries with comparatively lower emissions. As a result, credit-constrained carbon-intensive firms in these countries may consider moving their business to nations with laxer environmental restrictions, such as those in the BRIC countries. When emissions are transferred to these new areas as a result of this process, the original countries' overall CO2 emissions can be reduced.

In another scenario, when industries move to higher quantiles that represent countries with greater CO2 emissions, a different dynamic takes shape. Even though macroprudential rules

**Table 6. Quantile regression.**

| | Q10 | Q25 | Q50 | Q75 | Q90 |
|---|---|---|---|---|---|
| MaPP_3 | -0.020*** | 0.002 | 0.051*** | 0.080*** | 0.078*** |
| | (0.01) | (0.01) | (0.02) | (0.02) | (0.01) |
| trade | 0.015*** | 0.015*** | -0.011** | -0.025*** | -0.033*** |
| | (0.00) | (0.00) | (0.00) | (0.01) | (0.00) |
| fdi | -0.003 | -0.014 | 0.007 | 0.049 | 0.050 |
| | (0.01) | (0.02) | (0.04) | (0.05) | (0.03) |
| CPI | 0.008*** | 0.010*** | 0.006 | -0.005 | -0.003 |
| | (0.00) | (0.00) | (0.00) | (0.01) | (0.00) |
| ICT | -0.002*** | -0.002* | -0.003 | -0.001 | 0.000 |
| | (0.00) | (0.00) | (0.00) | (0.00) | (0.00) |
| GDP | -0.350*** | -0.368*** | -0.075 | -0.096 | 0.039 |
| | (0.02) | (0.05) | (0.07) | (0.10) | (0.06) |
| constant | 15.047*** | 15.191*** | 14.318*** | 16.556*** | 16.016*** |
| | (0.21) | (0.42) | (0.69) | (0.92) | (0.56) |
| N | 319 | 319 | 319 | 319 | 319 |

Standard errors in parentheses

* $p < 0.1$

** $p < 0.05$

*** $p < 0.01$

continue to restrict credit availability, the entry of carbon-intensive sectors from nations with lower CO2 emissions creates a new source of emissions. Due to laxer environmental rules, these businesses may flourish and produce more CO2. Therefore, the cumulative effect of carbon-intensive industries that shift their operations can affect the favourable impact that macroprudential policies have on emissions in these higher quantile countries.

In summary, the observed pattern of macroprudential policies exhibiting a negative effect on CO2 emissions in countries with lower emissions (10th quantile) and a positive effect in higher quantiles can be attributed to carbon leakage dynamics. The movement of industries across borders that is driven by varying regulatory environments can lead to intricate shifts in emissions patterns and subsequently impact the outcomes of macroprudential policies on CO2 emissions in different quantiles.

In essence, the observed sign reversal of the effect of macroprudential policies on CO2 emissions across quantiles could be tied to the intricate relationship between economic policies, global supply chains, and the potential for emissions to leak across borders.

## 4.4 Moderating effect of globalisation

Various levels of globalisation can lead to different outcomes in macroprudential policies. As shown in Table 7, the random effects model appeared to be the most suitable for the regression study. The interaction term (MaPP_3*KOF) demonstrated a significant negative coefficient, as outlined in Table 7. This implies that macroprudential policies will exert a stronger influence on CO2 emissions in an environment that is characterised by high levels of globalisation. This can be attributed to cultural and political globalisation. On one hand, globalisation has brought environmental awareness to the forefront, where various global green initiatives and civil society organisations have played a crucial role in heightening public consciousness about environmental issues. Consequently, the public tends to adopt more environmentally-friendly

**Table 7. Moderating effect.**

|  | ols | fe | re |
|---|---|---|---|
| MaPP_3 | 0.374*** | 0.084*** | 0.087*** |
|  | (0.06) | (0.01) | (0.01) |
| KOF | -0.008 | 0.010*** | 0.010*** |
|  | (0.01) | (0.00) | (0.00) |
| MaPP_3*KOF | -0.004*** | -0.001*** | -0.001*** |
|  | (0.00) | (0.00) | (0.00) |
| trade | -0.010** | -0.000 | -0.000 |
|  | (0.00) | (0.00) | (0.00) |
| fdi | -0.003 | 0.007* | 0.007* |
|  | (0.03) | (0.00) | (0.00) |
| CPI | 0.002 | 0.001*** | 0.001*** |
|  | (0.00) | (0.00) | (0.00) |
| ICT | -0.002 | -0.002*** | -0.002*** |
|  | (0.00) | (0.00) | (0.00) |
| GDP | 0.135 | 0.704*** | 0.690*** |
|  | (0.11) | (0.04) | (0.04) |
| constant | 13.243*** | 6.274*** | 6.418*** |
|  | (0.53) | (0.36) | (0.53) |
| N | 319 | 319 | 319 |
| r2 | 0.260 | 0.821 |  |
| r2_a | 0.241 | 0.810 |  |
| F | 13.650*** | 172.246*** |  |
| BP LM | chibar2(01) = 3676.26*** | | |
| chi2 |  |  | 1347.235*** |
| Hausman |  | chi2(7) = 12.72* | |

Standard errors in parentheses

\* $p < 0.1$

\*\* $p < 0.05$

\*\*\* $p < 0.01$

behaviours, which is a view that is similar to the work of [34]. This also means that the adoption of more environmentally-friendly behaviours will enhance the carbon reduction impact of greener investment decisions facilitated by macroprudential policies. However, on the other hand, with increased globalisation of political affairs, international collaborative organisations and environmental institutions are more likely to formulate rational international environmental agreements and regulations. These efforts, combined with macroprudential policies, can work towards achieving the SDGs.

## 4.5 Robust test

Four robustness tests were conducted to validate the primary findings. Initially, the sample was segmented into pre- and post-GFC periods to investigate the potential influence of external shocks on the nexus. In the second and third rounds of robust testing, the dependent variable and independent variable were systematically and individually replaced to assess the robustness of the basic regression model. Furthermore, additional control variables were introduced one by one to meticulously re-examine the underlying relationship.

**4.5.1 Robust test 1: Pre- and post-GFC periods.** An examination on the link between macroprudential policies and CO2 emissions was performed in both the pre-and post-GFC periods to acquire a thorough understanding of the findings. To achieve this, we investigated whether exogenous shocks have an impact on the link between macroprudential policy and CO2 emissions. In line with the work of [57, 58], the sample of this study was split into two time periods: before and after 2007, considering 2007 as the start of the GFC. The years preceding 2007 were referred to as the pre-GFC period, while the years from 2008 onward were regarded as the post-GFC period.

As shown in Table 8, the random effect model was most suitable for the regression study. There existed a negative correlation between macroprudential policies and CO2 emissions during the post-GFC period. Notably, it was intriguing to observe that macroprudential measures exhibited a positive influence on CO2 emissions in the pre-GFC period. The positive correlation observed during the pre-GFC period may be attributed to an environment where economic growth and development took precedence, potentially resulting in less stringent credit controls. This environment could lead to an uptick in industrial activities and energy consumption, subsequently contributing to higher CO2 emissions. However, in the post-GFC period, the global financial crisis prompted a stricter regulation of credit, which significantly impacted corporate economic activities by constraining access to funds. As a result, industrial activities were curtailed, leading to a reduction in CO2 emissions.

**Table 8. Robust test1.**

| | Pre-GFC | | | Post-GFC | | |
|---|---|---|---|---|---|---|
| | ols | fe | re | ols | fe | re |
| MaPP_3 | 0.078** | 0.018*** | 0.018*** | 0.068*** | -0.013*** | -0.013*** |
| | (0.03) | (0.00) | (0.00) | (0.01) | (0.00) | (0.00) |
| trade | -0.008* | 0.003*** | 0.003*** | -0.024*** | 0.000 | -0.000 |
| | (0.00) | (0.00) | (0.00) | (0.00) | (0.00) | (0.00) |
| fdi | 0.006 | -0.001 | -0.001 | -0.082* | 0.004 | 0.004 |
| | (0.03) | (0.00) | (0.00) | (0.05) | (0.00) | (0.00) |
| CPI | 0.007* | 0.002*** | 0.002*** | -0.005 | 0.001** | 0.001*** |
| | (0.00) | (0.00) | (0.00) | (0.00) | (0.00) | (0.00) |
| ICT | -0.002 | -0.001*** | -0.001*** | -0.006* | 0.001** | 0.002*** |
| | (0.00) | (0.00) | (0.00) | (0.00) | (0.00) | (0.00) |
| GDP | -0.081 | 0.490*** | 0.471*** | -0.074 | 0.267*** | 0.208** |
| | (0.07) | (0.06) | (0.05) | (0.09) | (0.10) | (0.10) |
| constant | 14.279*** | 8.796*** | 8.961*** | 16.648*** | 11.004*** | 11.581*** |
| | (0.58) | (0.50) | (0.64) | (1.11) | (0.97) | (1.00) |
| N | 176 | 176 | 176 | 143 | 143 | 143 |
| r2 | 0.099 | 0.809 | | 0.363 | 0.537 | |
| r2_a | 0.067 | 0.789 | | 0.335 | 0.479 | |
| F | 3.103*** | 111.936*** | | 12.921*** | 24.404*** | |
| BP LM | chibar2(01) = 1252.69*** | | | chibar2(01) = 596.12*** | | |
| chi2 | | | 673.912*** | | | 140.290*** |
| Hausman | chi2(5) = 4.50 | | | chi2(5) = 8.61 | | |

Standard errors in parentheses

* $p < 0.1$

** $p < 0.05$

*** $p < 0.01$

**Table 9. Robust test2.**

|  | ols | fe | re |
|---|---|---|---|
| map_r5 | 0.047*** | -0.005*** | -0.004*** |
|  | (0.01) | (0.00) | (0.00) |
| trade | -0.016*** | 0.001 | 0.001 |
|  | (0.00) | (0.00) | (0.00) |
| fdi | 0.002 | 0.013*** | 0.013*** |
|  | (0.03) | (0.00) | (0.00) |
| CPI | 0.001 | 0.001*** | 0.001*** |
|  | (0.00) | (0.00) | (0.00) |
| ICT | -0.003* | -0.003*** | -0.002*** |
|  | (0.00) | (0.00) | (0.00) |
| GDP | 0.005 | 0.997*** | 0.965*** |
|  | (0.05) | (0.04) | (0.05) |
| constant | 14.383*** | 4.130*** | 4.436*** |
|  | (0.50) | (0.42) | (0.55) |
| N | 297 | 297 | 297 |
| r2 | 0.235 | 0.735 |  |
| r2_a | 0.219 | 0.720 |  |
| F | 14.811*** | 129.614*** |  |
| BP LM | chibar2(01) = 2943.74*** | | |
| chi2 |  |  | 716.465*** |
| Hausman | chi2(5) = 21.95*** | | |

Standard errors in parentheses

\* $p < 0.1$

\*\* $p < 0.05$

\*\*\* $p < 0.01$

**4.5.2 Robust test 2: Replace independent variable.** Following the previous study of [40], a variable, map_r5, was constructed using a 5-year spanning window to aggregate the macroprudential tools, while the basic regression model utilised three-year rolling data. The result is reported in Table 9. The regression results show that the basic regression remained robust.

**4.5.3 Robust test 3: Replace dependent variable.** The model was re-estimated by replacing the dependent variable with CO2 emissions metric tons per capita (CE) and CO2 intensity (CI). In line with the study of [1, 2, 12, 34, 59, 60], the CO2 emissions metric tons per capita (CE) was chosen as one indicator of CO2 emissions. Furthermore, drawing inspiration from the research conducted by [39], CO2 emissions (kt) divided by land area (sq. km) was chosen to gauge CO2 intensity (CI) as another indicator of CO2 emissions. The results are reported in Table 10. The regression results show that the basic regression remained robust.

**4.5.4 Robust test 4: Add extra control variable.** The absence of relevant variables is likely to decrease the validity of empirical findings and bring about estimation basis. Therefore, a few factors were added to the model to see if there were changes in the main findings. Consistent with the findings of prior research [1, 2] which highlighted the significance of financial development and economic uncertainty in the context of CO2 emissions, financial development (FD) and economic uncertainty (EU) variables were incorporated into the analysis. FD was measured by real domestic credit to the private sector per capita. To capture EU, the global economic uncertainty index from https://worlduncertaintyindex.com/data/ was employed as the proxy of EU. Subsequently, these two variables were systematically introduced into the

**Table 10. Robust test3.**

|  | CI | | | CE | | |
|---|---|---|---|---|---|---|
|  | ols | fe | re | ols | fe | re |
| MaPP_3 | -0.044*** | -0.010*** | -0.010*** | 0.022*** | -0.010*** | -0.009*** |
|  | (0.01) | (0.00) | (0.00) | (0.01) | (0.00) | (0.00) |
| trade | -0.004 | -0.004*** | -0.004*** | 0.006*** | 0.000 | 0.000 |
|  | (0.00) | (0.00) | (0.00) | (0.00) | (0.00) | (0.00) |
| fdi | -0.073*** | 0.008* | 0.008* | -0.013 | 0.012*** | 0.012*** |
|  | (0.03) | (0.00) | (0.00) | (0.02) | (0.00) | (0.00) |
| CPI | 0.005** | -0.000 | -0.000 | -0.002 | 0.001 | 0.001 |
|  | (0.00) | (0.00) | (0.00) | (0.00) | (0.00) | (0.00) |
| ICT | 0.001 | -0.002*** | -0.002*** | -0.002** | -0.003*** | -0.003*** |
|  | (0.00) | (0.00) | (0.00) | (0.00) | (0.00) | (0.00) |
| GDP | 0.257*** | 0.722*** | 0.699*** | 0.575*** | 0.904*** | 0.874*** |
|  | (0.05) | (0.04) | (0.04) | (0.03) | (0.04) | (0.04) |
| constant | -1.541*** | -5.608*** | -5.397*** | -3.711*** | -6.785*** | -6.507*** |
|  | (0.46) | (0.36) | (0.41) | (0.26) | (0.34) | (0.37) |
| N | 319 | 319 | 319 | 319 | 319 | 319 |
| r2 | 0.249 | 0.554 |  | 0.652 | 0.714 |  |
| r2_a | 0.234 | 0.531 |  | 0.645 | 0.698 |  |
| F | 17.197*** | 62.566*** |  | 97.233*** | 125.418*** |  |
| BP LM | chibar2(01) = 3724.39*** | | | chibar2(01) = 3369.24*** | | |
| chi2 |  |  | 353.406*** |  |  | 740.305*** |
| Hausman |  | chi2(5) = 11.44** | |  | chi2(5) = 14.76** | |

Standard errors in parentheses

\* $p < 0.1$

\*\* $p < 0.05$

\*\*\* $p < 0.01$

regression model individually. The corresponding outcomes were then reported in Tables 11 and 12. The regression outcomes validated the robustness of the fundamental regression analysis.

## 5. Conclusion

### 5.1. Conclusion

This paper studied the impact of macroprudential policies on CO2 emissions over the period of 1992–2020 among G7 and BRIC countries. All in all, the results suggest a negative relationship between macroprudential policies and CO2 emissions. This phenomenon may arise from the impact of these policies on economic growth and investment decisions. When the relationship across various intervals along the CO2 emissions distribution was investigated, these policies were found to exacerbate CO2 emissions in countries that already had high levels of CO2 emissions due to carbon leakage. Globalisation was also found to strengthen the carbon reduction effect of macroprudential policies. When dividing the sample period into pre-GFC and post-GFC periods, it can be observed that the impact of macroprudential policies is consistent with the basic regression model during the post-GFC period. However, the positive relationship in the pre-GFC period is interesting to note. From this, the robust tests confirmed the basic regression results.

**Table 11. Robust test4.1: Adding control variable EU.**

| | ols | fe | re |
|---|---|---|---|
| MaPP_3 | 0.065*** | -0.006*** | -0.004* |
| | (0.01) | (0.00) | (0.00) |
| trade | -0.015*** | 0.002** | 0.002** |
| | (0.00) | (0.00) | (0.00) |
| fdi | 0.009 | 0.009** | 0.009* |
| | (0.03) | (0.00) | (0.00) |
| CPI | 0.002 | 0.002*** | 0.002*** |
| | (0.00) | (0.00) | (0.00) |
| ICT | -0.001 | -0.002*** | -0.002*** |
| | (0.00) | (0.00) | (0.00) |
| GDP | -0.010 | 0.920*** | 0.828*** |
| | (0.05) | (0.04) | (0.04) |
| EU | -1.860*** | -0.293*** | -0.317*** |
| | (0.34) | (0.05) | (0.06) |
| constant | 14.498*** | 4.824*** | 5.682*** |
| | (0.45) | (0.35) | (0.43) |
| N | 319 | 319 | 319 |
| r2 | 0.264 | 0.778 | |
| r2_a | 0.248 | 0.766 | |
| F | 15.973*** | 151.004*** | |
| BP LM | chibar2(01) = 3065.44*** | | |
| chi2 | | | 755.664*** |
| Hausman | | chi2(6) = 73.81*** | |

Standard errors in parentheses

* $p < 0.1$

** $p < 0.05$

*** $p < 0.01$

## 5.2. Policy recommendations

The results of this study have important implications for policy in G7 and BRIC countries. On one hand, policymakers should integrate policies that prioritise sustainable economic growth and environmental conservation. This involves aligning macroprudential policies with environmental objectives to ensure that economic activities do not lead to excessive CO2 emissions. By implementing stringent credit controls that encourage investments in renewable energy, energy efficiency, and clean technologies, a country can achieve a balance between economic prosperity and environmental protection. On the other hand, it is important to strengthen and enforce environmental regulations to prevent carbon leakage caused by industries that relocate to countries with weaker environmental standards. This involves setting stringent emission targets, implementing green certifications, and monitoring the environmental practices of industries to discourage them from transferring emissions to less regulated regions. Finally, the popularity of greener technologies, the lowering of trade barriers, increasing environmental awareness and international cooperation should all be taken into account by policymakers as supplemental actions to support and enhance the effects of macroprudential policies on environmental sustainability.

**Table 12. Robust test4.2: Adding control variable FD.**

| | ols | fe | re |
|---|---|---|---|
| MaPP_3 | 0.049*** | -0.008*** | -0.006*** |
| | (0.01) | (0.00) | (0.00) |
| trade | 0.001 | 0.001 | 0.001 |
| | (0.00) | (0.00) | (0.00) |
| fdi | -0.013 | 0.012*** | 0.011** |
| | (0.02) | (0.00) | (0.00) |
| CPI | -0.002 | 0.002*** | 0.002*** |
| | (0.00) | (0.00) | (0.00) |
| ICT | -0.001 | -0.003*** | -0.002*** |
| | (0.00) | (0.00) | (0.00) |
| GDP | -0.412*** | 0.976*** | 0.899*** |
| | (0.06) | (0.04) | (0.04) |
| FD | 0.013*** | -0.001** | -0.001 |
| | (0.00) | (0.00) | (0.00) |
| constant | 16.544*** | 4.378*** | 5.085*** |
| | (0.44) | (0.37) | (0.45) |
| N | 319 | 319 | 319 |
| r2 | 0.418 | 0.756 | |
| r2_a | 0.405 | 0.743 | |
| F | 31.968*** | 133.476*** | |
| BP LM | chibar2(01) = 2517.11*** | | |
| chi2 | | | 713.284*** |
| Hausman | chi2(6) = 62.01*** | | |

Standard errors in parentheses

* p < 0.1

** p < 0.05

*** p < 0.01

### 5.3. Limitation

The study is only a preliminary examination of how macroprudential policies in G7 and BRIC nations can affect CO2 emissions. This study still has limits in the following areas. First, the sample only covered top 11 GDP-ranked countries. Although there is no dispute regarding the selection criteria for economies, focusing on a larger group of industrialised economies, such as the G-20, may provide the research community with broader-ranging conclusions and recommendations. Second, researchers could extend their investigations to explore the impact of different types tightening, easing, capital-related, liquidity-related, and asset-based macroprudential policies on environmental degradation. Finally, it is important to consider various sources of carbon emissions, such as coal, oil, and gas.

## Supporting information

**S1 Data.**
(XLSX)

## Author Contributions

**Conceptualization:** Heng Luo.

**Data curation:** Heng Luo.

**Formal analysis:** Heng Luo.

**Investigation:** Heng Luo.

**Methodology:** Heng Luo.

**Project administration:** Heng Luo.

**Resources:** Heng Luo.

**Software:** Heng Luo.

**Supervision:** Fakarudin Kamarudin.

**Validation:** Heng Luo.

**Visualization:** Heng Luo.

**Writing – original draft:** Heng Luo.

**Writing – review & editing:** Heng Luo.

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
