## [Decision Letter · Decision Letter 0]

24 Sep 2023

PONE-D-23-26694Macroprudential Policies and CO2 emissions: A Comparative Analysis of G7 and BRICS CountriesPLOS ONE

Dear Dr. LUO,

Thank you for submitting your manuscript to PLOS ONE. After careful consideration, we feel that it has merit but does not fully meet PLOS ONE’s publication criteria as it currently stands. Therefore, we invite you to submit a revised version of the manuscript that addresses the points raised during the review process.

ACADEMIC EDITOR:Please access the reviewer's comments carefully. Authors should be consistent in the use of BRICS or BRIC. I will suggest you stick to the former. If you have to go by BRIC implying the omission of South Africa due to the 11 countries panel as sample study, please provide substantial justifications for such. The contributions should be strengthened more. There only two which are not substantial enough. For instance, that you are the first to conduct this study is not sufficient as contributions. How does this study add to the issues of CO2 escalation in the countries of study. How does your study relates and promotes of the SDGs?   

We look forward to receiving your revised manuscript.

Kind regards,

Ridwan Lanre Ibrahim

Academic Editor

PLOS ONE

Reviewers' comments:

Reviewer's Responses to Questions

**Comments to the Author**

1. Is the manuscript technically sound, and do the data support the conclusions?

Reviewer #1: Partly

2. Has the statistical analysis been performed appropriately and rigorously? 

Reviewer #1: Yes

3. Have the authors made all data underlying the findings in their manuscript fully available?

Reviewer #1: Yes

4. Is the manuscript presented in an intelligible fashion and written in standard English?

Reviewer #1: No

5. Review Comments to the Author

Reviewer #1: The title of this paper is "Macroprudential Policies and CO2 Emissions: A Comparative Analysis of G7 and BRICS Countries." It is a research paper, and I have several suggestions that the authors may consider to improve it:

Abstract section:

The language in the abstract is generally clear, but it can be further improved for coherence. I suggest adding research insights to the abstract.

Introduction section:

The structure is somewhat confusing, and the organization of information can be clearer. I recommend refining the division of paragraphs and the organization of content. Additionally, some sentences are overly long, so please ensure readability. The transition from citations to the literature review is not very clear.

Literature Review section:

1.The structure is somewhat disorganized, and the connection between different studies and viewpoints is not very smooth. I suggest reorganizing paragraphs to present the themes and findings of previous research more clearly.

2.The literature review mentions various factors influencing CO2 emissions but does not provide specific research findings or controversies. The citation format should adhere to academic standards.

3.Some previous research conclusions are mentioned, but specific examples or data to support these conclusions are lacking. To increase credibility, you can cite actual results or data from studies.

4.The last sentence mentions the impact of macroprudential policies on CO2 emissions but does not explicitly connect this viewpoint to the research question.

5.The literature review content is limited; more comprehensive coverage is needed.

Data and Methodology section:

1.There are significant language expression issues, please focus on revising them.

2.Additionally, when describing the reasons for selecting these countries, you can provide more explanation to illustrate why these countries were chosen (based on GDP rankings) and how they are relevant to the research question.

3.You can briefly explain why you chose the time period from 1992 to 2020 and why the MaPR_3 index requires the use of 3-year rolling data.

4.Mentioned the use of some country-level control variables, but did not provide detailed explanations for their roles or reasons.

5.The BASIC results section provides regression results but lacks sufficient explanation.

6.While mentioning the control variables (ICT, foreign direct investment, trade, GDP, CPI), there is not enough explanation to justify why these variables were chosen and their relationship with CO2 emissions.

7.The Quantile regression section mentions results at different percentiles but does not provide enough explanation to clarify why the results vary across different percentiles.

Conclusion section:

Overall, it is relatively clear, but when discussing future research directions, you can be more specific in your descriptions.

6. PLOS authors have the option to publish the peer review history of their article (what does this mean?). If published, this will include your full peer review and any attached files.

Reviewer #1: No

---

## [Author Response · Author response to Decision Letter 0]

19 Oct 2023

PLOS ONE

Dear Editors and Reviewers:

Thank you for your letter and the reviewers’ comments concerning our manuscript (PONE-D-23-26694). Those comments are all valuable and very helpful for revising and improving our paper, as well as the important guiding significance to our research. We have studied comments carefully and have made corrections which we hope meet with approval. Revised portions are marked up using the “Track Changes” function in the paper. The main corrections in the paper and the responses to the editor’s and reviewer’s comments are as follows:

Response to editor’s Comments: 

Point 1: Please access the reviewer's comments carefully. Authors should be consistent in the use of BRICS or BRIC. I will suggest you stick to the former. If you have to go by BRIC implying the omission of South Africa due to the 11 countries panel as sample study, please provide substantial justifications for such. 

Response 1: We greatly appreciate your comments and assure you that we will carefully address each one. It was our mistake to use BRICS, and we would like to correct it to BRIC. There are several reasons for this change. Firstly, economist Jim O'Neill has argued that South Africa does not belong in BRICS due to its relatively small economy (source: https://www.theguardian.com/world/2013/mar/24/south-africa-bric-developing-economy). Secondly, South Africa is the most recent member to join BRICS, having become a member in December 2010. Therefore, the sample period including South Africa as part of BRICS only covers 11 (2020-2010+1) out of 29(2020-1992+1) years. Additionally, South Africa accounts for only 1.8% of the total BRICS population (source: https://www.statssa.gov.za/?p=11355). 

Additionally, we have rewritten factors that influence the sample selection in line73-line89 in the manuscript. 

Furthermore, in the work of Pata (2021), the link between renewable energy and ecological footprint is only examined in BRIC countries. Taking all of these factors into consideration, we have decided to use BRIC in our analysis.

Point 2: The contributions should be strengthened more. There only two which are not substantial enough. For instance, that you are the first to conduct this study is not sufficient as contributions. How does this study add to the issues of CO2 escalation in the countries of study. How does your study relates and promotes of the SDGs? 

Response 2: We gratefully appreciate your comment. Regarding the contributions and how to add to the issues of CO2 escalation as well as relates and promotes of the SDGs, we have rewritten this part (reflected in line98-line118) and we added a moderating variable in our regression as another contribution in section 4.4(line411-line430) in the manuscript. Thanks for your valuable comments.

 

Response to Reviewer 1 Comments:

Point 1: Abstract section:

The language in the abstract is generally clear, but it can be further improved for coherence. I suggest adding research insights to the abstract.

Response 1: We appreciate your valuable comments. We have sent this paper for editing. According to the reviewer’s suggestions, we have rewritten the content in line20-line33 in the manuscript.

Point 2: Introduction section:

The structure is somewhat confusing, and the organization of information can be clearer. I recommend refining the division of paragraphs and the organization of content. Additionally, some sentences are overly long, so please ensure readability. The transition from citations to the literature review is not very clear.

Response 2: We gratefully appreciate your comment. We have sent this paper for editing and reorganized the introduction section. The revised part is reflected in line38-line121.Thanks for your valuable comments 

Point 3.1: Literature Review section:

The structure is somewhat disorganized, and the connection between different studies and viewpoints is not very smooth. I suggest reorganizing paragraphs to present the themes and findings of previous research more clearly.

Response 3.1: We gratefully appreciate your comment. We have reorganized the literature review section. The revised part is reflected in line124-line168.Thanks for your valuable comments.

Point 3.2: Literature Review section:

The literature review mentions various factors influencing CO2 emissions but does not provide specific research findings or controversies. The citation format should adhere to academic standards.

Response 3.2: We greatly appreciate your comment. We have incorporated discussions regarding the controversies surrounding the impact of macroprudential policies on environmental quality in lines 134 to 166. The citation format has also been revised accordingly.

Point 3.3: Literature Review section:

Some previous research conclusions are mentioned, but specific examples or data to support these conclusions are lacking. To increase credibility, you can cite actual results or data from studies.

Response 3.3: We gratefully appreciate your comment. We have cited the results related to this part in lines 134 to 158.

Point 3.4: Literature Review section:

The last sentence mentions the impact of macroprudential policies on CO2 emissions but does not explicitly connect this viewpoint to the research question.

Response 3.4: We greatly appreciate your comment. Macroprudential policies can influence environmental quality through their impact on credit or investment. Further details on this matter can be found in lines 134 to 158.

Point 3.5: Literature Review section:

The literature review content is limited; more comprehensive coverage is needed.

Response 3.5: Thank you very much for your valuable comments. We have taken your suggestion into account and thoroughly revised the Literature Review section, addressing the points you raised. We would also like to note that despite our best efforts, we encountered a scarcity of relevant literature on this topic in Google Scholar. We have made sure to include the most pertinent sources available. We have rewritten this part in line123-line168.

Point 4.1: Data and Methodology section:

There are significant language expression issues, please focus on revising them.

Response 4.1: We gratefully appreciate your comment. We have sent this paper for editing already.

Point 4.2: Data and Methodology section:

Additionally, when describing the reasons for selecting these countries, you can provide more explanation to illustrate why these countries were chosen (based on GDP rankings) and how they are relevant to the research question.

Response 4.2: We gratefully appreciate for your comment. We have added more contents for the sample selection in line73-line89.

Point 4.3: Data and Methodology section:

You can briefly explain why you chose the time period from 1992 to 2020 and why the MaPR_3 index requires the use of 3-year rolling data.

Response 4.3: We gratefully appreciate for your comment. We add the expiations in line206-line210 for the first question and in line243-line256 for the second question.

Point 4.4: Data and Methodology section:

Mentioned the use of some country-level control variables, but did not provide detailed explanations for their roles or reasons.

Response 4.4: We gratefully appreciate for your comment. We have added explanations for selecting these control variables in line267-line277.

Point 4.5: Data and Methodology section:

The BASIC results section provides regression results but lacks sufficient explanation.

Response 4.5: It is really true as Reviewer suggested. We have added explanations to enhance the power of persuasion in line328-line354.

Point 4.6: Data and Methodology section:

While mentioning the control variables (ICT, foreign direct investment, trade, GDP, CPI), there is not enough explanation to justify why these variables were chosen and their relationship with CO2 emissions.

Response 4.6: We appreciate for your valuable suggestions. Thus, we have added explanations for selecting these control variables in line267-line277.

Point 4.7: Data and Methodology section:

The Quantile regression section mentions results at different percentiles but does not provide enough explanation to clarify why the results vary across different percentiles.

Response 4.7: A detailed explanation of the quantile regression result analysis is written in the section of “4.3. Quantile regression” and concluded in line379-line407. Thanks for your valuable comments. 

Point 5: Conclusion section:

Overall, it is relatively clear, but when discussing future research directions, you can be more specific in your descriptions.

Response 5: Considering the Reviewer’s suggestion, we have added future research in line525 -line534. Thanks for your valuable comments.

 

Reference

1. Pata UK. Linking renewable energy, globalization, agriculture, CO2 emissions and ecological footprint in BRIC countries: A sustainability perspective. Renew Energy. 2021;173: 197–208. doi:10.1016/j.renene.2021.03.125

---

## [Decision Letter · Decision Letter 1]

24 Nov 2023

PONE-D-23-26694R1Macroprudential policies and CO2 emissions: A comparative analysis of G7 and BRIC countriesPLOS ONE

Dear Dr. LUO,

Thank you for submitting your manuscript to PLOS ONE. After careful consideration, we feel that it has merit but does not fully meet PLOS ONE’s publication criteria as it currently stands. Therefore, we invite you to submit a revised version of the manuscript that addresses the points raised during the review process. Comments to author:

As you can see your paper near acceptance, you are kindly advise to ensure diligence in doing the current revision. Take caution not to include details that will require more verification or a new review process as much as we encourage improvements on the manuscript at all time. 

We look forward to receiving your revised manuscript.

Kind regards,

Ridwan Lanre Ibrahim

Academic Editor

PLOS ONE

Journal Requirements:

Reviewers' comments:

Reviewer's Responses to Questions

**Comments to the Author**

1. If the authors have adequately addressed your comments raised in a previous round of review and you feel that this manuscript is now acceptable for publication, you may indicate that here to bypass the “Comments to the Author” section, enter your conflict of interest statement in the “Confidential to Editor” section, and submit your "Accept" recommendation.

Reviewer #1: All comments have been addressed

2. Is the manuscript technically sound, and do the data support the conclusions?

Reviewer #1: Yes

3. Has the statistical analysis been performed appropriately and rigorously? 

Reviewer #1: Yes

4. Have the authors made all data underlying the findings in their manuscript fully available?

Reviewer #1: Yes

5. Is the manuscript presented in an intelligible fashion and written in standard English?

Reviewer #1: No

6. Review Comments to the Author

Reviewer #1: Overall the revisions are good, but the language suggests further embellishment to improve the logic and readability of the whole text.

7. PLOS authors have the option to publish the peer review history of their article (what does this mean?). If published, this will include your full peer review and any attached files.

Reviewer #1: No

---

## [Author Response · Author response to Decision Letter 1]

26 Nov 2023

PLOS ONE

Dear Editors and Reviewers:

Thank you for your letter and the reviewers’ comments concerning our manuscript (PONE-D-23-26694R1). Those comments are all valuable and very helpful for revising and improving our paper, as well as the important guiding significance to our research. We have studied comments carefully and have made corrections which we hope meet with approval. Revised portions are marked up using the “Track Changes” function in the paper. The main corrections in the paper and the responses to the editor’s and reviewer’s comments are as follows:

Response to Reviewer 1 Comments:

Point 1: Overall the revisions are good, but the language suggests further embellishment to improve the logic and readability of the whole text.

Response 1: We appreciate your valuable comments. The paper has been submitted for further editing, and we have also restructured our paragraphs for clarity and coherence.

---

## [Editor Report · Decision Letter 2]

11 Dec 2023

Macroprudential policies and CO2 emissions: A comparative analysis of G7 and BRIC countries

PONE-D-23-26694R2

Dear Dr.Luo,

We’re pleased to inform you that your manuscript has been judged scientifically suitable for publication and will be formally accepted for publication once it meets all outstanding technical requirements.

Kind regards,

Ridwan Lanre Ibrahim

Academic Editor

PLOS ONE
---

## [Editor Report · Acceptance letter]

28 Dec 2023

PONE-D-23-26694R2 

PLOS ONE

Dear Dr. Luo, 

I'm pleased to inform you that your manuscript has been deemed suitable for publication in PLOS ONE. Congratulations! Your manuscript is now being handed over to our production team.

Kind regards, 

on behalf of

Professor Ridwan Lanre Ibrahim 

Academic Editor

PLOS ONE